# Efficient PDE Solutions using Hartley Neural Operators in Physics-Informed Networks: Potentials and Limitations

## Abstract

In this work, we introduce novel differentiable architectures for solving partial differential equations (PDEs) using well-known Discrete Hartley Transform. We incorporate Hartley Neural Operators (HNO) into Physics-Informed Neural Operator Networks (PINOs). Our analysis concentrates on two pivotal PDEs: 1. the one-dimensional diffusion equation, which holds significance not only in machine learning but also across a broad spectrum of physical sciences and engineering disciplines; and 2. the one-dimensional Thermodynamic Energy Equation that is commonly used in weather data analysis. Our implementation of HNO that employ real-valued linear transforms into the PINO architecture results in significant run time improvements. We show that reconstruction loss is lower than other recently introduced operators that may be used for the above PDEs. Importantly, we find that HNOs naturally satisfy the governing physical laws and equations specific to the PDEs under consideration. However, our empirical observations suggest that the benefits of HNO diminishes in certain scenarios where the underlying physical conditions at the boundary are less tractable and involve complex numbers. As an example of a potential failure mode we illustrate that in the case of the one-dimensional Burger's equation, traditional Fourier Neural Operators outperform their Hartley counterparts. Our results indicate that a combination of neural operators including Fourier and Hartley transforms may be better to effectively address the specific type, and/or context of physical problem at hand.

## 1 Introduction

**Designing a real-valued Neural Operator for Deep Learning.** Fourier Neural Operators (FNOs) Li et al. (2020), combines Fourier analysis with neural networks to study multi-scale phenomena. The conventional methodology employed in utilizing neural networks for solving differential equations often entails the process of discretizing the complete domain into a grid and subsequently predicting values at each individual grid point. Nevertheless, the scalability of this approach deteriorates significantly as the problem dimensionality grows. The approach of Fourier Neural Operators involves operating directly in the Fourier domain, leveraging its inherent capacity to handle differential operators. This methodology results in solutions that are both more salable and accurate.

While revolutionary, the FNO framework has certain limitations. For example, consider using Fourier Transforms to modify neural network representations to solve partial differential equations (PDEs) with real-valued variables, and solutions that are known a priori to be real valued also. In such a setting, it is possible that the resulting scheme provides suboptimal solutions unless an excessively precise resolution is used to eliminate imaginary components appropriately. The Hartley Transform Hartley (1942), like the Fourier Transform, can transform signals between the time (or spatial) and frequency domains with the key advantage is that it produces real-valued output for real-valued input. Therefore, it operates fully over the space of real numbers – so post processing conversions are not necessary. This is advantageous for real-valued PDEs because it eliminates the need to use complex numbers inherent to the Fourier Transform. This distinction has proven advantageous in the fields of image processing and the solution of certain partial differential equations in which the existence of real-valued solutions is guaranteed.

We introduce the concept of Hartley Neural Operators (HNO), which follows the same inspiration and framework as FNO while leveraging the advantages of Hartley Transform. While the concept of applyng the Hartley Transform to neural operators has, to our knowledge, not been done before, the concept of leveraging Hartley transforms for deep learning is nothing new Arenas & Serrano-Gotarredona (1996). Our study, however, incorporates HNO into neural operator learning frameworks. Specifically, we apply various combinations of HNO and FNO layers in a neural network to compare solutions of a small sample set of PDEs that are of interest to machine learning and the physical sciences alike. The first method combined the HNO and FNO layers to evaluate their potency and collaboration. The second procedure replaced all four layers of the structure with HNO layers. After rigorous evaluations, both approaches improved performance measures. These changes shed light on the potential of HNO within the PINO framework and lay the foundation for further investigation.

**Our Contributions.** Our key contribution is that we modify and adapt the FNO frameworks using Hartley Transforms as a drop-in replacement for the Fourier Transform within the neural operator layer architecture. These Hartley Neural Operators are explored for their potential use and efficacy in this organized framework. We also use a cyclic convolution method, instead of a standard element-wise multiplication for the HNO layers, which efficiently handles periodic signals and is computationally advantageous when using the Discrete Hartley Transform (DHT), as it maintains a consistent signal length without extending boundaries. Otherwise, the convolution theorem for the Hartley Transform is much more complex than a simple multiplication in the transformed basis.

## 2 NEURAL HARTLEY TRANSFORM

**Preliminaries.** The *continuous* Hartley transformation of a function $f(t)$ is mathematically defined by the following integral transform:

$$H_f(\omega) = \int_{-\infty}^{\infty} f(t) \cdot \operatorname{cas}(\omega t) \, dt \tag{1}$$

where the cas function is defined as the sum of cosine and sine functions involving frequency $\omega$, that is, $\operatorname{cas}(\omega t) = \cos(\omega t) + \sin(\omega t)$. Similar to the Fourier transform, the Hartley transform has the unique property that it is involutionary, that is, the inverse is defined by:

$$f(t) = \int_{-\infty}^{\infty} H_f(\omega) \cdot \operatorname{cas}(\omega t) \, d\omega \tag{2}$$

For transforming discrete signals such as vectors, we discretize the integral in equation 2 as follows. Given a sequence $f[n]$ of length $N$, the *Discrete* Hartley Transform (DHT) Bracewell (1983) is defined as:

$$H_k = \sum_{n=0}^{N-1} f[n] \cdot \left( \cos\left(\frac{2\pi kn}{N}\right) + \sin\left(\frac{2\pi kn}{N}\right) \right) \tag{3}$$

The Discrete Hartley Transform (DHT) is a linear operation that involves the multiplication of a vector $(x_0, \cdots, x_{N-1}) \in \mathbb{R}^N$ by an $N \times N$ matrix. In order to obtain the initial value of $x_n$ from $H_k$, it suffices to perform the Discrete Hartley Transform (DHT) on $H_k$ and afterwards normalize the output by a factor of $(1/N)$. Thus, the DHT can be considered a self-inverse structure, but incorporating a normalization/scaling factor.

Though the convolution of the DHT is not as straightforward and elegant as the discrete Fourier transform (DFT), it is still fairly computationally inexpensive. For the cyclic convolution of vectors, $x$ and $y$ producing $z$ (all length $N$), post-DHT operations become simpler. If $X, Y$, and $Z$ represent the DHTs of $x, y$, and $z$ respectively, then $Z$ can be obtained as,

$$Z_k = \frac{X_k(Y_k + Y_{N-k}) + X_{N-k}(Y_k - Y_{N-k})}{2}$$

$$Z_{N-k} = \frac{X_{N-k}(Y_k + Y_{N-k}) - X_k(Y_k - Y_{N-k})}{2}. \tag{4}$$

**Similarities to Wavelet transforms and Discrete Cosine Transforms.** Integral transforms such as the wavelet transform (Grossmann & Morlet, Daubechies (1988)), the Hartley transform, and

the discrete cosine transform (DCT) Ahmed et al. (1974), analyze and represent data in multiple domains. Both use basis functions to analyze and represent signals. For multi-resolution signal analysis, the wavelet transform derives basis functions from a "mother wavelet" by dilation and translation Akansu & Haddad (1992). Whereas sinusoidal and cosinusoidal functions are used in the Hartley transform. Meanwhile, the DCT transforms a signal largely utilizing cosine functions, making it particularly effective at concentrating signal energy into a few coefficients. DCT is frequently used in image and signal processing, especially for JPEG compression Wallace (1991). However, DCT has limitations. It may introduce artifacts, specifically "blocking artifacts", when used in high-compression scenarios. Due to its inability to portray crisp transitions or edges, DCT reduces visual detail. Furthermore, its fixed basis functions are not adaptable, which can impede optimal signal representation in a variety of settings.

Both wavelet and Hartley transformation formulas utilize integrals of the original signal multiplied by a kernel (the wavelet function Kaiser (1994) for wavelet transforms and the cas function for Hartley transforms, respectively). The transforms enable practitioners to decompose the frequency or scale components of a signal, thereby revealing its underlying properties. Signal processing requires switching between time (or spatial) and frequency (or scale) domains, making these techniques necessary. Although versatile in storing signals across scales, wavelet transforms have limitations. Choosing the optimal wavelet function necessitates domain-specific knowledge, making the decision difficult. A suboptimal selection can reduce the precision of signal representation. Shift variance and edge effects can also present difficulties. While wavelets excel at capturing transient events, they may fall short when it comes to recording sinusoidal signals or phase information. Due to its real-valued structure and coupled sinusoidal and cosinusoidal functions, the representations provided by Hartley transform are often significantly more robust. Thus, many applications that prioritize computational performance and clarity may benefit more from the Hartley transform than DCT or wavelet transforms.

**Combining HNO with Physics Informed Neural Networks.** Physics-Informed Neural Networks (PINNs) for nonlinear PDEs are recently explored for various reasons Raissi et al. (2019). Compared to traditional neural networks, PINNs offer better adherence to the laws of physics, efficiency, and versatility. These naturally led to the development of Physics-Informed Neural Operator Networks (PINOs) Li et al. (2023), which amalgamate physics principles into neural operator structures, proving invaluable in areas like fluid dynamics and environmental modeling. The utilization of the Hartley transform in PINO systems has the potential to enhance computational efficiency and strengthen learning mechanisms. A recent study Rosofsky et al. (2023) introduces a framework for integrating partial differential equations (PDEs), essential for understanding and simulating many physical occurrences. The framework includes initial data generation, boundary conditions, and physics-informed neural operators. Their solutions demonstrate accuracy comparable to the literature. A primarily focus of theirs is the one-dimensional Burgers' equation Burgers (1948) - a fundamental partial differential equation that encompasses the interplay between non-linear advection and diffusion, which can be written as:

$$\frac{\partial u}{\partial t} + u\frac{\partial u}{\partial x} = \nu\frac{\partial^2 u}{\partial x^2} \tag{5}$$

Here, we apply the Hartley Neural Operator (HNO) to the one-dimensional Burgers' equation for comparison to their work. We also apply the HNO framework to two other PDEs, which are traditionally real-valued.

**Operating on the Diffusion Equation, a key equation in Machine Learning.** First, the diffusion equation Fick (1855), often also known as the heat equation, describes how heat, concentration, and population density spread through time and space. In Machine Learning applications, PDEs are used in various ways. For example, Laplacian Eigenmaps smooth or interpolate high-dimensional data sets using diffusion equation-like constructions **?**. Label propagation in graph-based semi-supervised learning uses the diffusion equation **?**. Modeling the 'flow' of data points in a feature space helps uncover anomalies and is used in time-series analysis **?**. Understanding stochastic gradient descent and other optimization techniques can be done using diffusion equation concepts **?**. The diffusion equation is also extremely versatile in physics due to its being key to understanding heat transmission between states of matter. In one dimension, this PDE can be written as,

$$\frac{\partial u}{\partial t} = D\frac{\partial^2 u}{\partial x^2} \tag{6}$$

**Applications to PDEs Useful In Weather Forecasting.** Second, the thermodynamic energy equation Gill (1982) with advection terms and assuming hydrostatic balance - a fundamental concept that links gravity with vertical pressure gradient force - that simplifies the equation by focusing on vertical forces, which is a good approximation for most meteorological uses. For our choice of diabatic heating term, we chose a linear function in $T$ to simplify matters.

$$\frac{\partial T}{\partial t} + w\frac{\partial T}{\partial z} = Q_0 T + Q_1 \tag{7}$$

**Defining Losses.** We primarily the methodology used in setting up FNO architectures to our losses:

1. $\mathcal{L}_{L2}$: This loss aligns the model predictions with the training data. It's calculated as the relative mean squared error (MSE) between the training data and the model outputs, normalized by the norm of the actual values.

2. $\mathcal{L}_{reconstruction}$: This loss ensures the model's adherence to known physical laws, like PDEs or conservation laws. Defined as the MSE between the breach of the physical principle and its ideal value, which here for practicality is the MSE between the residuals and zero, indicating the physics fit exactly.

Finally, define the Total Loss function as the weighted sum of the loss functions described above:

$$w_{L2} \times \mathcal{L}_{L2} + w_{reconstruction} \times \mathcal{L}_{reconstruction} \tag{8}$$

Unlike the previous study, we ignore initial condition losses (since their GitHub implementation has this commented out in their code anyway). We use unity weights for these runs. Both their studies and ours use experimental weight selection for the loss function rather than hyperparameter optimization, though our choice of unity is more arbitrary than anything.

**Using GRF For Initial Conditions.** In the construction of initial conditions, similar to the aforementioned study, Gaussian Random Fields (GRFs) are utilized Gudder (1978). GRFs are a type of spatially correlated random processes that have played a crucial role in generating realistic initial conditions for diverse simulations, particularly in the fields of geostatistics and cosmology. The Matérn function Matérn (1986) is widely used as a covariance function in Gaussian random fields (GRFs) because of its flexibility to represent various degrees of smoothness in the data. Through the utilization of the Matérn function within a Gaussian Random Field (GRF) Tilmann Gneiting & Schlather (2010), it is possible to create spatially consistent formations that possess distinct attributes dictated by the parameters of the function. The Matérn function is fundamentally responsible for introducing a predetermined degree of smoothness and correlation length to the resulting field, hence facilitating a deliberate synthesis of spatial data. When employed for the purpose of establishing initial conditions, this methodology guarantees that the resulting data exhibits statistical qualities that closely correspond to real-world observations.

**Defining Hartley Transforms using PyTorch Fast Fourier Transforms.** Our investigation required Fourier transforms to depict the Hartley transform since PyTorch lacks native Hartley Transform capabilities. The Hartley transform (H(u)) is closely related to the Fourier transform (F(u)): $H(u) = \text{Re}[F(u)] - \text{Im}[F(u)]$, where $\text{Re}$ and $\text{Im}$ represent the real and imaginary components of the Fourier transform. The equation was crucial to our analytical and computational work. We used PyTorch tools to adapt a Numpy version Panizza (2021) and enhance its multidimensional functionality.

## 3 EXPERIMENTS

**Setting Up Our Neural Network Architecture to Parallel Recent Work for Comparison.** Our research followed the same structure as Rosofsky et al. (2023). The model is characterized by its widths, modes, and total layers, with a strong emphasis on Gaussian error linear units (GELUs) as the activation function Hendrycks & Gimpel (2023). GELUs are preferred in physics-informed deep learning because they have a non-zero second derivative, making ReLUs less suited. In our 1-D configurations, the FNO structure includes four layers with widths [16, 24, 24, 32] and modes

|  | $\mathcal{L}_{L2}$ | $\mathcal{L}_{reconstruction}$ | $\mathcal{L}_{TOTAL}$ |
|---|---|---|---|
| HNO | 0.6099 | 0.3726 | 0.9825 |
| 2 HNO, 2 FNO | 0.0292 | 0.0255 | 0.0547 |
| 1 HNO, 3 FNO | 0.0217 | 0.0169 | 0.0386 |
| FNO | 0.0240 | 0.0255 | 0.0495 |

Table 1: Comparison of L2 and reconstruction losses between architectures using combinations of the standard Fourier Neural Operator method (FNO) and our Hartley Neural Operator (HNO) substitution.

[15, 12, 9, 9]. The design produces a 128-width, completely linked layer. We optimized our model on PyTorch using the Adam method with parameters $\beta_1 = 0.9$ and $\beta_2 = 0.999$, across 500 epochs, with an initial learning rate of 0.001.

**Is it beneficial to use Swish as an Activation Function in PINO?** Strategic use of GELUs combines ReLU with sigmoid functions to provide non-linearity and handle the vanishing gradient issue in PDE situations. However, another important activation function in this area is Swish, created by Google Brain Ramachandran et al. (2017), which combines input and sigmoid. Swish overcomes constraints via ReLU and LeakyReLU. Both GELUs and Swish produce smoother function outputs, although Swish's simple architecture may be more computationally efficient. Their performance depends on the application. GELUs are recommended for physics-informed jobs in Fourier Neural Operator frameworks, and Swish is versatile. Problem specifics, computational limits, and empirical outcomes determine the final choice.

**Using RK4 with Finite Difference to Obtain Exact Solutions.** The Runge-Kutta order 4 (RK4) method, a popular numerical integration method, was used to solve partial differential equations in our work (just as Rosofsky et al. (2023) did in their study) in order to have "ground truth" for what the exact solutions to the PDEs were to compare the predictions of the neural operator networks we employed. The complexity of PDEs, which incorporate several independent variables like time and space, required a two-pronged approach: We first used finite difference methods to discretize the spatial components of the PDE, turning it into a time-dependent ODE system. The RK4 approach then was used to integrate the system across time after spatial discretization. Given a 1-dimensional spatial domain, we set the domain boundaries as $xmin = 0$ and $xmax = 1$. The domain is discretized into $Nx = 100$ spatial grid points, which provides the spatial resolution of the simulation. The time step size for advancing the solution is selected as $dt = 1 \times 10^{-3}$, which determines the temporal resolution and influences the stability of the algorithm. The simulation is run until an end time $tend = 1.0$, where we choose these parameters to ensure both stability and efficiency of the numerical method.

## 4 RESULTS

**A Hybrid HNO/FNO Approach Gives the Best Results for Burgers'.** The first experiment, comparisons of the loss values for different architectures of PINO run to solve the one dimensional Burgers' equation is shown in Table 1. The outcomes of the losses related to several variations of PINO, which incorporate varying combinations of Fourier and Hartley layers in their 4-layer architecture, are presented in Table 1. The table provides a systematic comparison of the effectiveness of these configurations, bringing valuable insights into the influence of various layer combinations on the performance of the network. Each row in the table represents a distinct variation of the PINO algorithm, indicating the specific number of Fourier and Hartley layers that have been included. The losses associated with each model, which are potentially presented in the following columns, provide a quantifiable assessment of their correctness or error rate. The Hartley design with four layers has the most unfavorable loss values among the four variations, whereas the architecture including a single Hartley layer followed by three Fourier layers demonstrates the most favorable loss outcomes. Furthermore, as depicted in Figure 1, the comparison between the exact solution (obtained using a fourth-order Runge-Kutta solver) and the solution predicted by the PINOs with partial Hartley and partial Fourier layers exhibits a remarkable level of agreement. Notably, the absolute error remains consistently low as the solution evolves over time.

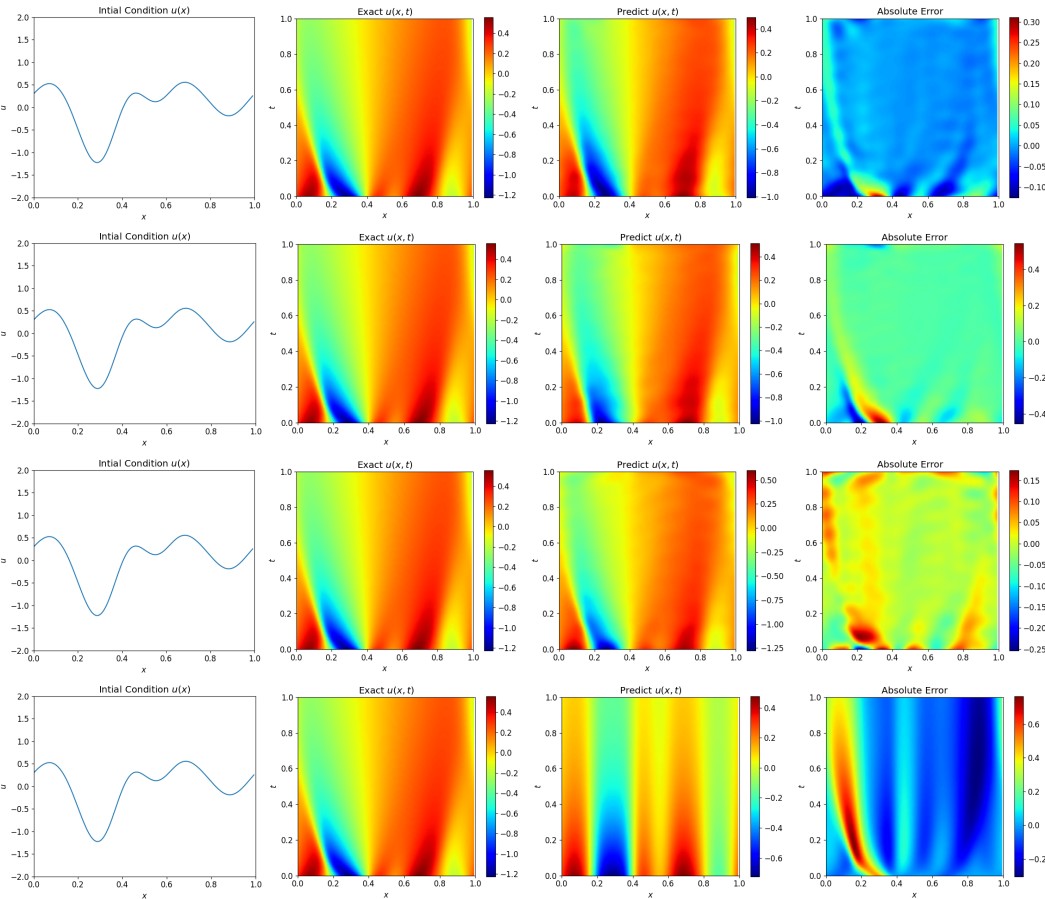

Figure 1: Comparison of Exact vs. Predicted (with absolute error shown) of the 1-D Burger's equation with the given initial condition and periodic boundary conditions using 1.) Top Panel: FNO with 4 layerss; 2.) Middle Top Panel: 1 HNO layer and 3 FNO layers; 2.) Middle Bottom Panel: 2 HNO layers and 2 FNO layers; 4.) Bottom Panel: HNO with 4 layers.

| | activation | FNO $\mathcal{L}_{L2}$ | $\mathcal{L}_{reconstruction}$ | HNO $\mathcal{L}_{L2}$ | $\mathcal{L}_{reconstruction}$ |
|---|---|---|---|---|---|
| Heat/Diffusion Eq | GELU | 0.00126 | 0.00216 | 0.00385 | 0.00214 |
| Heat/Diffusion Eq | Swish | 0.00116 | 0.00216 | 0.00468 | 0.00214 |
| TDE, Linear Q(T) | GELU | 0.00457 | 0.01085 | 0.04044 | 0.01069 |
| TDE, Linear Q(T) | Swish | 0.00472 | 0.01086 | 0.04047 | 0.01069 |

Table 2: Comparison of L2 and Function error between the standard Fourier Neural Operator method (FNO) and our Hartley Neural Operator (HNO) substitution.

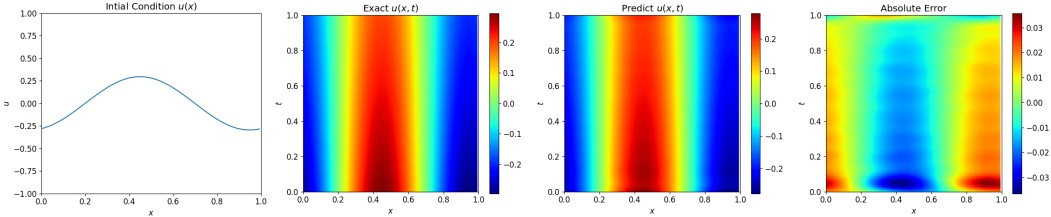

Figure 2: 1-D diffusion equation results with GELU activation function using a 4-layer FNO architecture.

**Four-Layer HNO Improves Reconstruction Loss for Solving the Diffusion Equation and TDE.**
Table 2 displays the comparative reconstruction and L2 losses for the FNO and HNO structures, using GELU and Swish activation functions. One noteworthy finding derived from the table is the comparatively reduced reconstruction loss exhibited by the HNO model. This outcome implies a more advanced level of alignment with the fundamental principles of the Diffusion and TDE equations. Nevertheless, the aforementioned observation is accompanied with an elevation in L2 losses for the HNO, suggesting a somewhat reduced level of agreement with the empirical data.

**FNO Causes Periodic Error, HNO Low Random Error in Diffusion**. Figures 2 and 3 offer significant insights into the performance of the FNO and HNO designs when employed for the resolution of the Diffusion equation. The evaluation of these structures is conducted under varying initial conditions, utilizing randomized Gaussian Random Field (GRF) fields. In the context of the 4-layer FNO, it is worth mentioning that the discrepancy between the precise and projected observations displays a recurring pattern, a periodicity that diminishes over time, whereas in the case of the 4-layer HNO, the error is comparatively reduced and demonstrates a greater degree of randomness as the solution progresses through time.

**For TDE with Linear Q(T) FNO Again Causes Periodic Error, while HNO's Error is Non-periodic, Both Increase as the Solution Evolves in Time.** In addressing the TDE equation, as illustrated in Figure 4 and Figure 5, we offer a thorough perspective on the outcomes obtained through the inclusion of a linear diabatic heating component and the utilization of identical randomized Gaussian random field (GRF) beginning conditions. It is noteworthy that the absolute error between the exact and predicted observations exhibits a similar periodicity when the 4-layer FNO

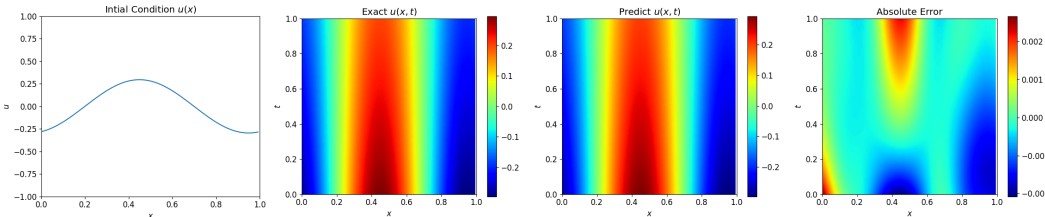

Figure 3: 1-D diffusion equation results with GELU activation function using a 4-layer HNO architecture.

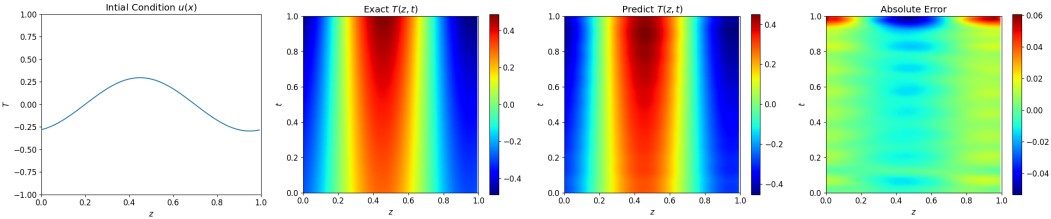

Figure 4: 1-D TDE equation, with linear Q results with GELU activation function using a 4-layer FNO architecture.

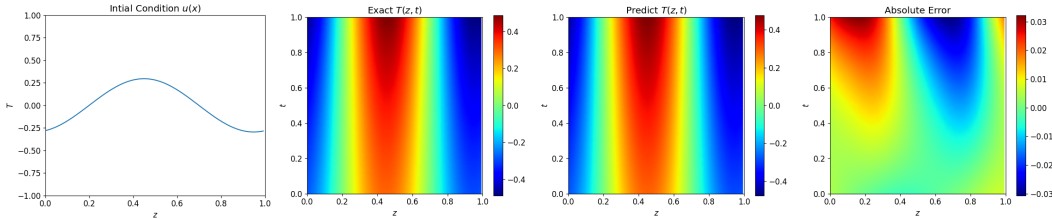

Figure 5: 1-D TDE equation, with linear Q results with GELU activation function using a 4-layer HNO architecture.

architecture is employed, akin to the case of the diffusion equation. However, this periodicity is not as pronounced as the solution progresses in time. Conversely, when the 4-layer HNO is utilized, the absolute error displays less periodicity than the FNO case, is an order of magnitude less than the FNO case, and displays less randomness than the HNO case in the 1-D diffusion equation. In fact, the error initially remains relatively low, but as the solution evolves over time, the error begins to increase.

**No Appreciable Difference in Activation Function Choice.** As shown in table 2, there was no appreciable difference in the overall neural network performance when using Swish over GELU, despite being self-gating and having been shown to work better for certain neural network architectures designed to solve PDEs.

## 5    DISCUSSION

We undertook a comprehensive adaptation of the PINO framework, originally proposed by Rosof-sky et al. (2022). Our study's chief aim was to explore the integration of Hartley Neural Operators (HNO) within this structure. Upon examining the Burgers' equation, we found that using HNO re-sulted in a heightened overall loss compared to its FNO equivalent. Notably, a combined approach employing both HNO and FNO layers displayed superior performance and a much better fit to the exact solution, then Rosofsky et al. (2023). Further, our exploration of the Diffusion equation and the TDE equation, using a linear depiction of the diabatic heating component, revealed subtle dif-ferences between the two neural operators. The HNO showed a lesser reconstructive loss compared to the FNO, but there was a minor rise in L2 loss for both.

When analyzing the time-dependent evolution of absolute errors in the diffusion equation, the FNO four-layer architecture revealed a distinct periodic error trend, whereas the HNO four-layer architec-ture revealed a more random error trajectory, albeit with an order of magnitude lower discrepancy. The observed periodicity in FNO error may suggest that the FNO architecture fails to account for specific data characteristics, such as boundary conditions or periodic patterns. The presence of os-cillatory error patterns in the Partial Differential Equation (PDE) suggests that the FNO architecture may be responsive to specific solution frequencies. These results may be ascribed to resonance phenomena, in which specific frequencies of the partial differential equation's solutions are either amplified or attenuated.

An analogous discrepancy was noted in the context of the TDE PDE with regard to the FNO. The HNO four-layer architecture initially matched the correct solution, but with time, prediction and ground truth diverged. The model's inability to effectively describe the underlying dynamics of the partial differential equation (PDE), especially over longer time periods, may explain these disparities. The observed divergence and larger L2 loss values may be due to overfitting, numerical instability, temporal discretization issues, boundary condition errors, or TDE complications with the selection of a linear diabatic heating term. In our scenario, where a randomized Gaussian random field (GRF) with a Matérn covariancThat said, however, the HNO still outperformed the FNO in the TDE solution prediction.

Finally, the choice of activation function between GELU and Swish did not lead to any substantial differences between either the HNO or FNO four-layer architectures and was not applied to the hybrid architecture. That said since we were working strictly in one-dimension and using simplified versions of the PDEs under investigation, solving the fully-fledged PDEs in three-dimensions may still benefit from such a choice in activation function.

## 6    CONCLUSION & FUTURE WORK

In conclusion, our research compellingly underscores the profound implications of melding real-valued transforms, with particular emphasis on the Hartley Transform, into the architecture of neural operators when grappling with streamlined primitive equations. By embarking on this innovative path, we have illuminated a pioneering methodology adept at decoding pivotal meteorological PDEs, such as the Thermodynamic Heat Equation. The merger of the Hartley Transform with these equations not only allows for a more nuanced understanding of atmospheric dynamics but also bestows upon our models an enhanced level of fidelity, ensuring predictions that resonate closely with real-world weather phenomena. The juxtaposition of our findings from the 1-D Diffusion and Thermodynamic Energy Equations with the Burgers' equation offers a holistic perspective. The comparative results accentuate that domain-specific PDEs, especially those inherently real-valued in their solutions, align more harmoniously with real-valued transforms like the Hartley. In stark contrast, equations like the Burgers', which might exhibit complex components or behaviors in their solutions, reveal the nuances and potential limitations of a purely real-valued approach.

It is also clear that additional ablation experiments that isolate and analyze specific neural operator elements or arrangements within the model architecture can help us understand their effects. An in-depth investigation of the training process, particularly the periodic error patterns, may be beneficial. In a similar manner, boundary conditions, temporal discretizations, and brain architecture may explain model efficacy differences. The quality and representativeness of the training data must also be closely examined, taking into account long-term discrepancies. Different activation functions, dropout rates, and batch normalization could also be examined in the neural operator's structure. The main goal of this ablation research is to identify specific sources of absolute error and high L2 loss while improving the model's design and training methods to reduce reconstruction loss and better fit the exact, numerical solutions.

Future research should aim to further explore the PINO framework, particularly in three dimensions. Further expanding the utilization to a wider range of intricate partial differential equations (PDEs), with particular emphasis on the extensive collection of fundamental equations that are vital to numerical weather prediction is also of interest. It would be beneficial, therefore, to employ a Hartley Neural Operator network to process weather model data with varying resolutions. For example, in recent studies, it has been demonstrated that Deep Learning models, such as FourCastNet Pathak et al. (2022), exhibit efficacy as viable options for NWP with the utilization of the Adaptive Fourier Neural Operator (AFNO) attention mechanism, have the capability to make accurate forecasts with low-resolution data with up to a maximum of seven days in advance. Future research, which is already ongoing, will strive to answer the question of whether or not can it do better if we apply the HNO framework modifications and create an AHNO attention mechanism instead.

As the research landscape continues to evolve, we anticipate that additional applications of the Hartley transform within PINOs specifically, and Neural Operator learning in general will materialize, contributing innovative resolutions to a diverse set of scientific and engineering challenges.

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

## A  APPENDIX

### A.1  HNO Skeleton

The Hartley Neural Operator applies the operations in algorithm 1 and then follows the same skeleton as Rosofsky et al. (2023), the code for which can be found at https://github.com/shawnrosofsky/PINO_Applications, except that anywhere there is a Fast Fourier Transform operation taken, we drop in our DHT from Algorithm 1.

### A.2  Additional Figures from results

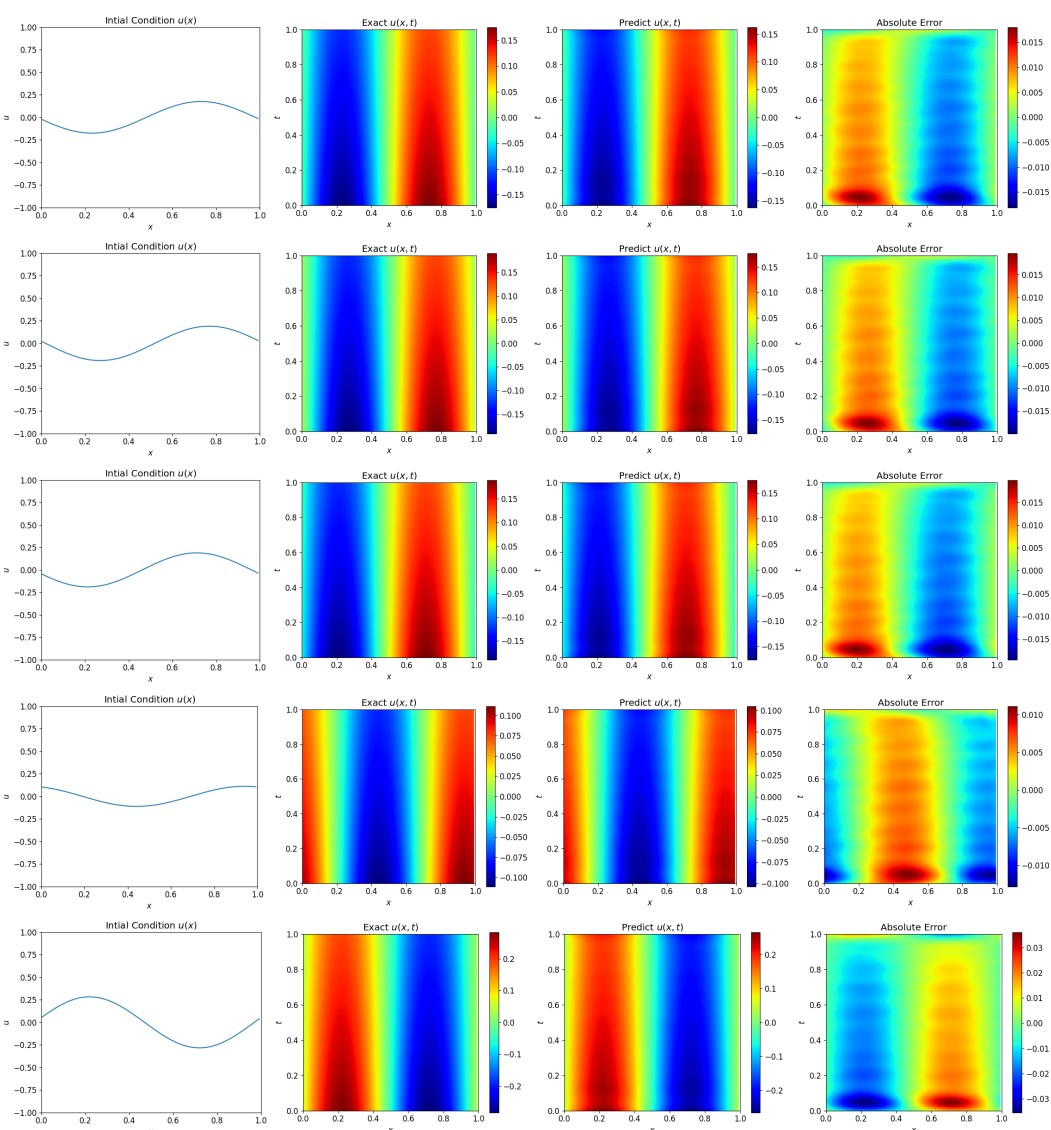

Figure 6: Additional 1-D diffusion equation results with GELU activation function using a 4-layer FNO architecture.

---

**Algorithm 1** Discrete Hartley Transform and Complex Multiplication

---

1: **function** DHT($x$)
2:     $X \leftarrow$ Compute FFT of $x$
3:     **return** Real part of $X$ minus Imaginary part of $X$
4: **end function**
5: **function** IDHT($X$)
6:     $dims \leftarrow$ Dimensions of $X$
7:     $n \leftarrow$ Product of all $dims$
8:     $X \leftarrow$ DHT($X$)
9:     **return** $X$ divided by $n$ element-wise
10: **end function**
11: **function** COMPL_MUL1D($x, y$)
12:     $X \leftarrow$ DHT($x$)
13:     $Y \leftarrow$ DHT($y$)
14:     $X_{\text{flip}} \leftarrow$ Flip and roll $x$
15:     $Y_{\text{flip}} \leftarrow$ Flip and roll $y$
16:     $Y_{\text{plus}} \leftarrow Y + Y_{\text{flip}}$
17:     $Y_{\text{minus}} \leftarrow Y - Y_{\text{flip}}$
18:     $Z \leftarrow 0.5 \times$ (Element-wise multiply $X$ with $Y_{\text{plus}}$ and $X_{\text{flip}}$ with $Y_{\text{minus}}$)
19:     **return** IDHT(Z)
20: **end function**

---

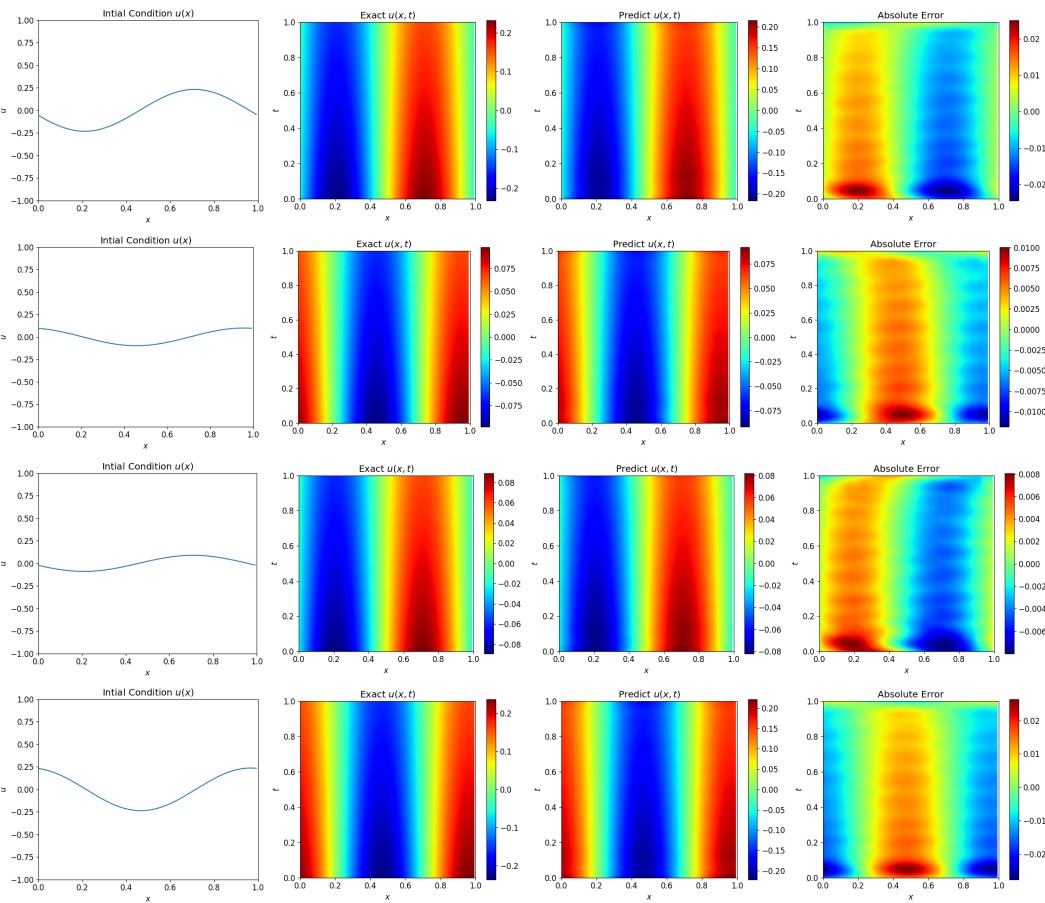

Figure 7: Additional 1-D diffusion equation results with GELU activation function using a 4-layer FNO architecture.

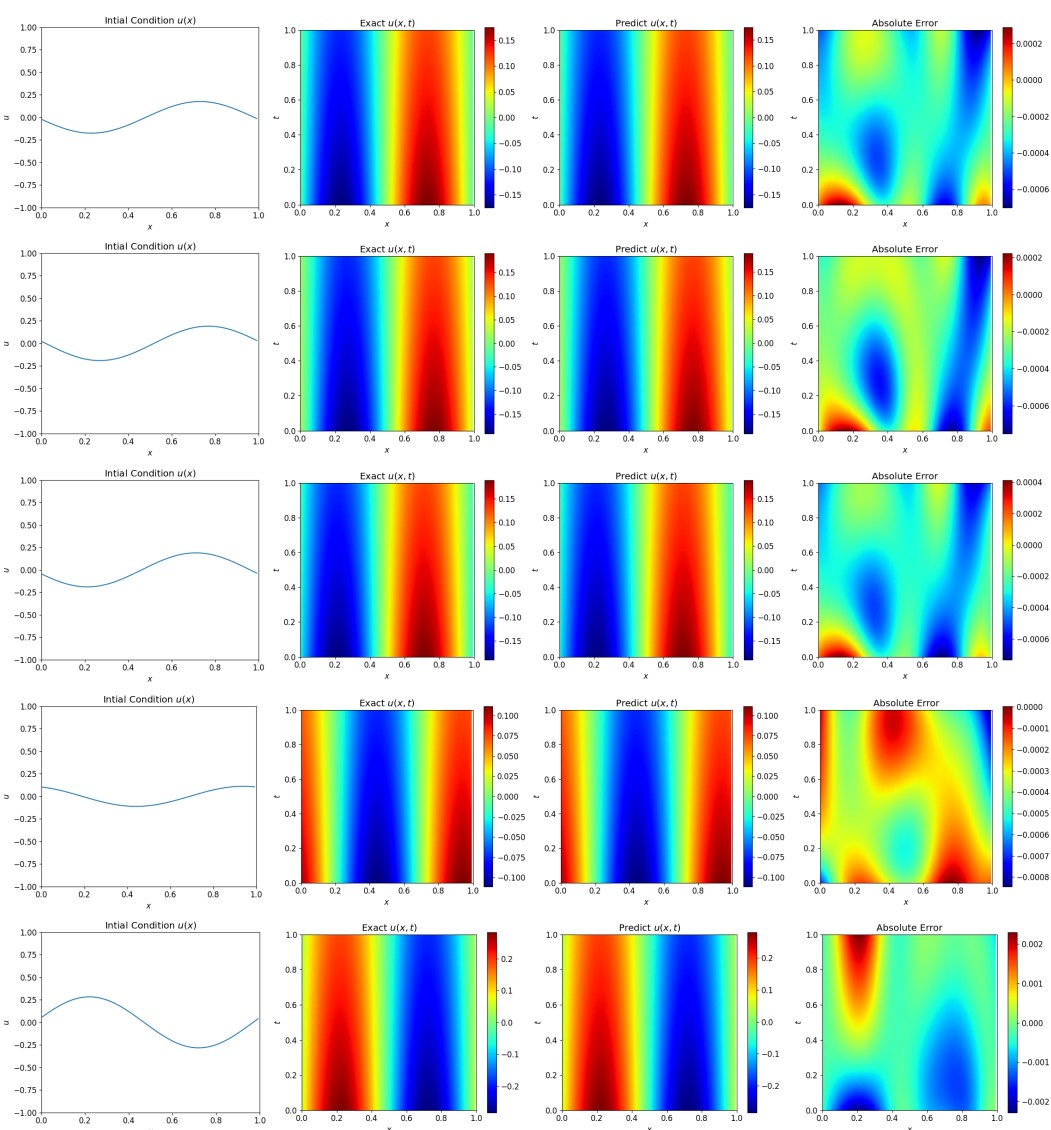

Figure 8: Additional 1-D diffusion equation results with GELU activation function using a 4-layer HNO architecture.

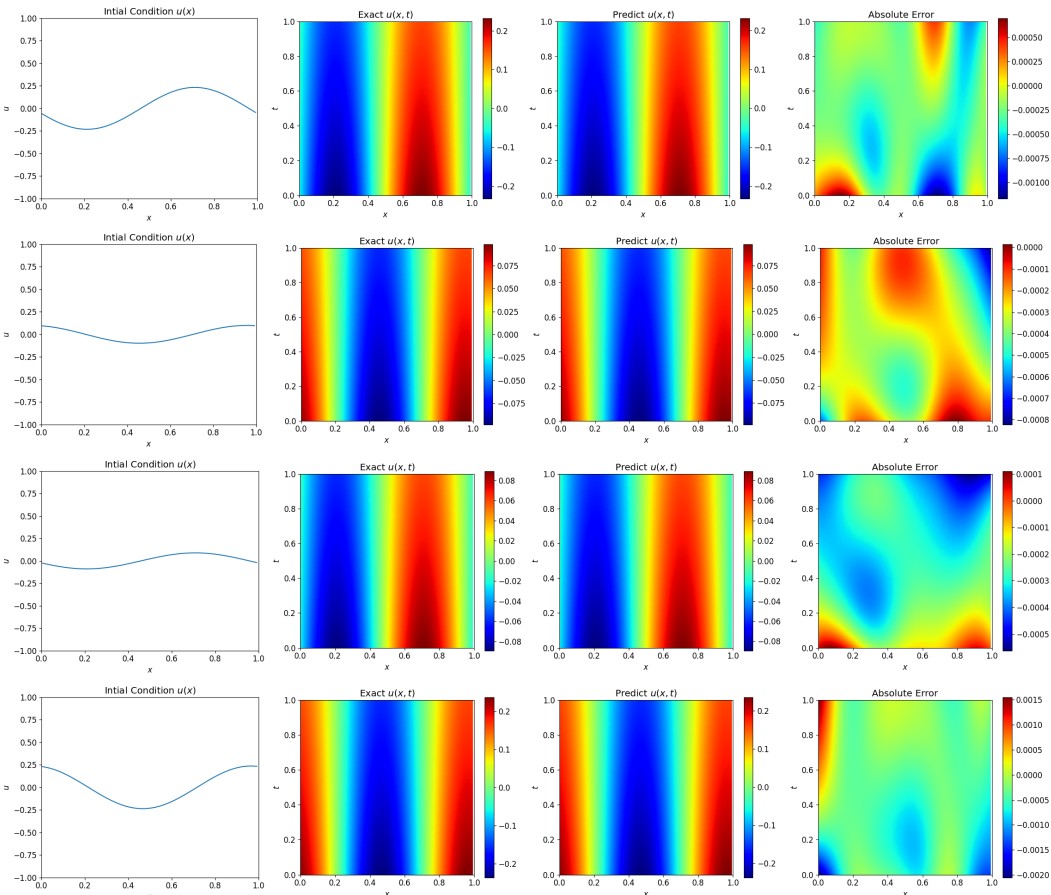

Figure 9: Additional 1-D diffusion equation results with GELU activation function using a 4-layer HNO architecture.

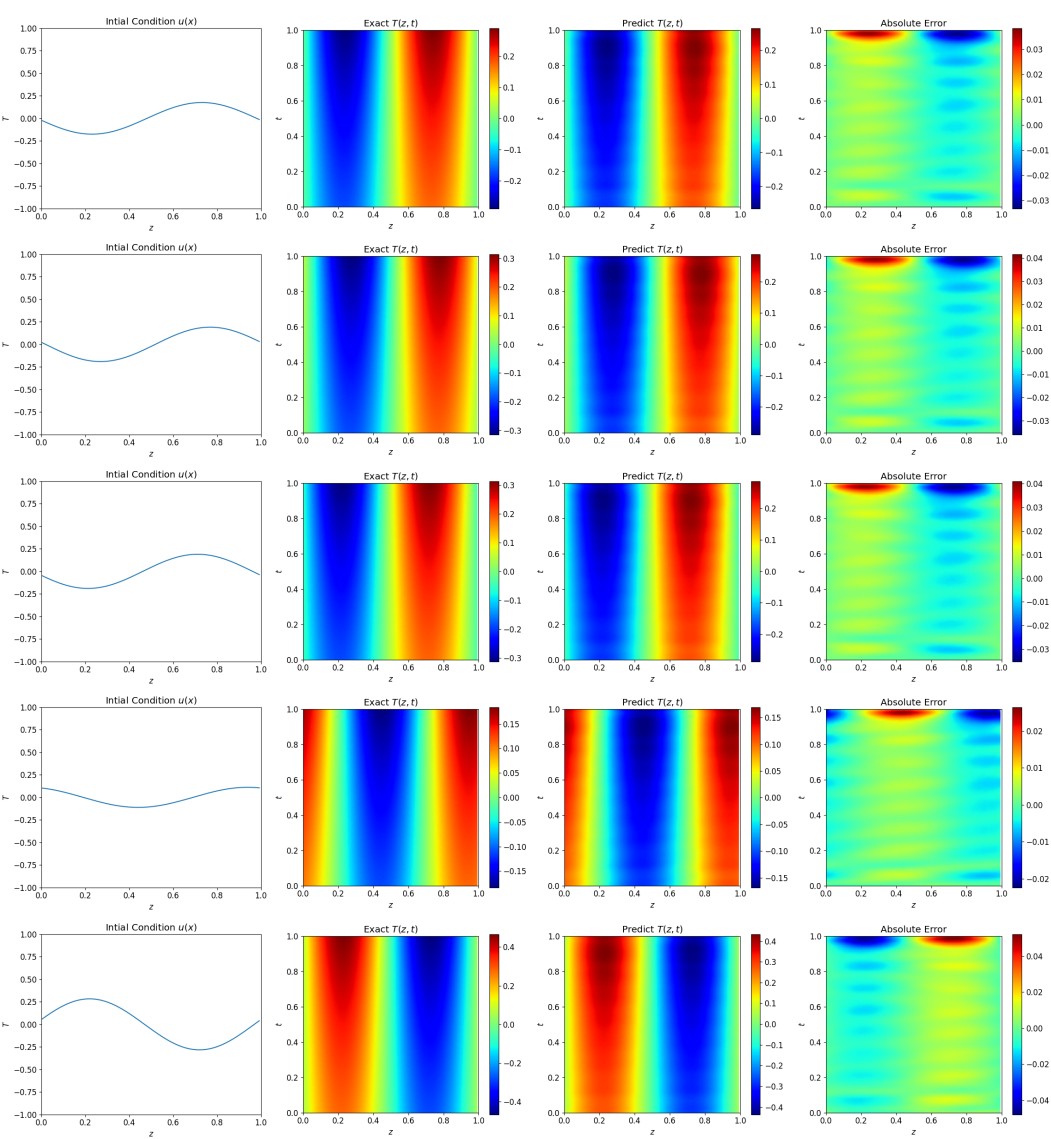

Figure 10: Additional 1-D TDE equation, with linear Q results with GELU activation function using a 4-layer FNO architecture.

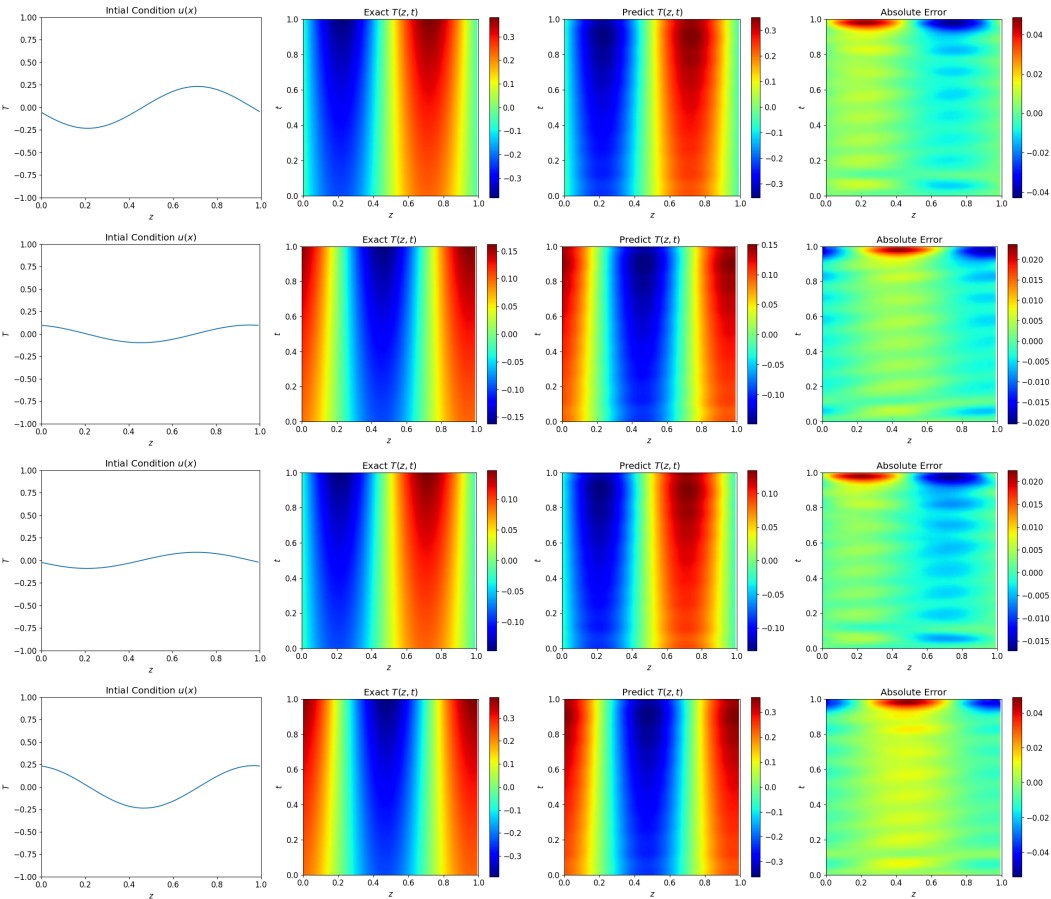

Figure 11: Additional 1-D TDE equation, with linear Q results with GELU activation function using a 4-layer FNO architecture.

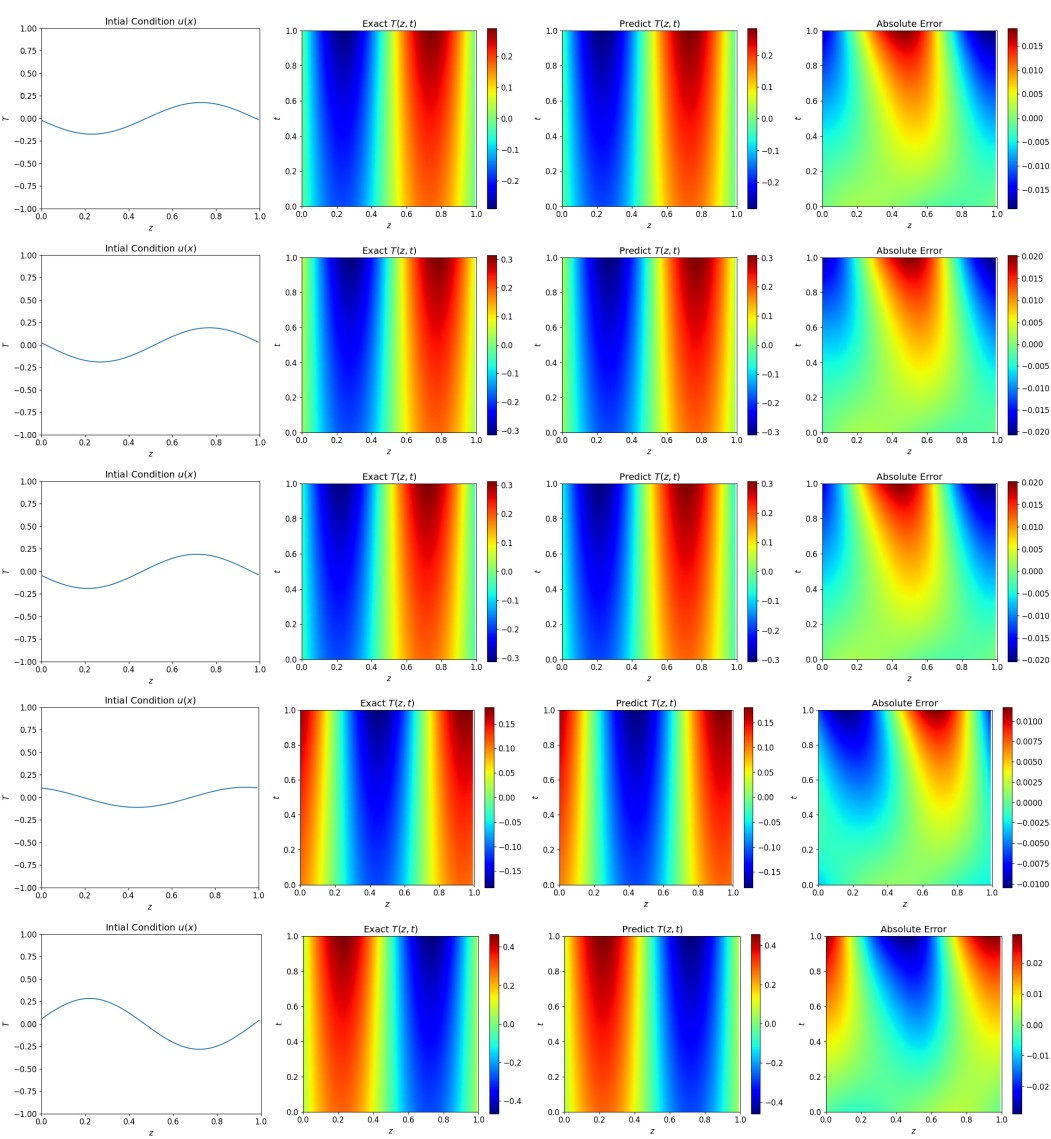

Figure 12: Additional 1-D TDE equation, with linear Q results with GELU activation function using a 4-layer HNO architecture.

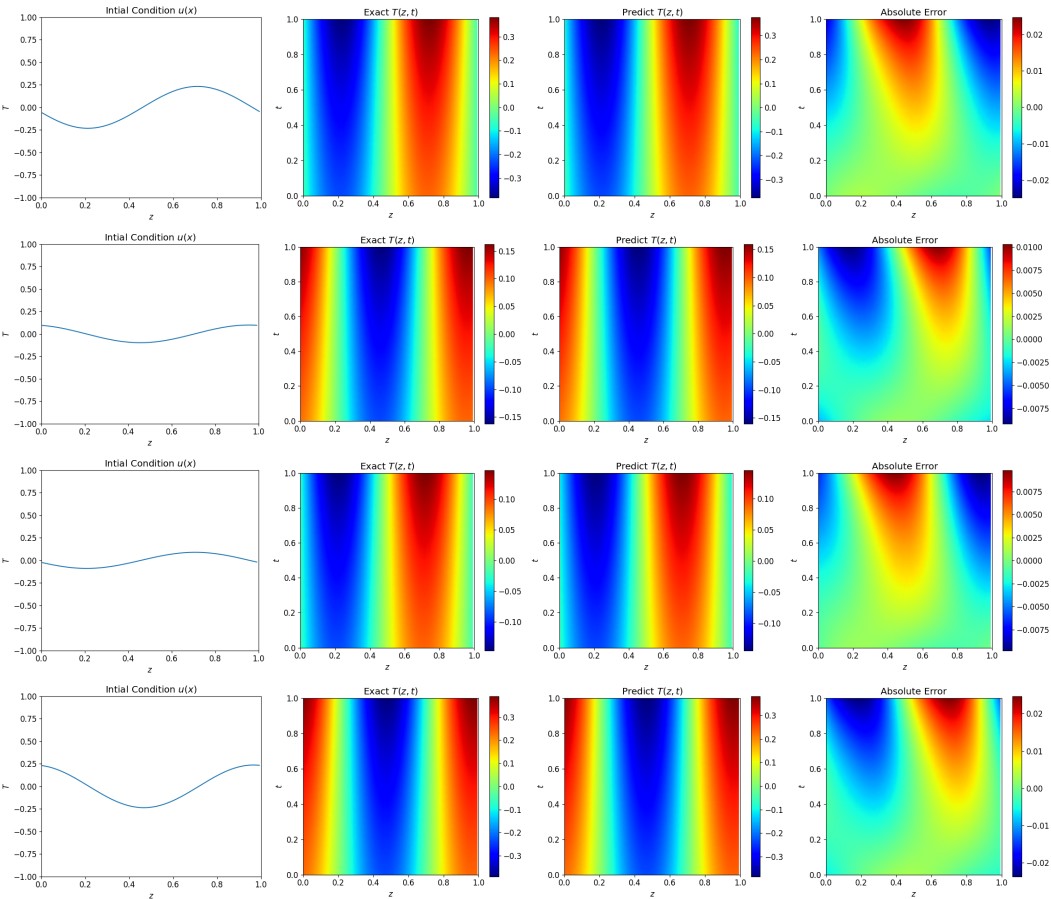

Figure 13: Additional 1-D TDE equation, with linear Q results with GELU activation function using a 4-layer HNO architecture.

