# OpenReview forum: "Efficient PDE Solutions using Hartley Neural Operators in Physics-Informed Networks: Potentials and Limitations"
_ICLR.cc/2024/Conference — ICLR 2024 Conference Withdrawn Submission_

### Official Review · Reviewer_cVFx · 2023-10-27

**Soundness:** 1 poor
**Presentation:** 2 fair
**Contribution:** 2 fair
**Rating:** 3
**Confidence:** 4

**Summary:**

This work considers the problem of solving partial differential equations with a physics-informed neural networks (PINNs) approach. The authors propose a novel neural operator architecture, based on the Discrete Hartley Transform, as an alternative to the popular Fourier neural operator, which is based on the Fast Fourier Transform. The main advantage of the Hartley transform is that it allows for a fast convolution (similar to the FFT) but enforces the resulting function to be real as opposed to complex-valued. In a second part of the paper, the authors evaluate the architecture on two partial differential equations: Burger's equation and a diffusion equation.

**Strengths:**

- The approach of the authors that incorporate the Hartley transform into neural operators appear to be novel.
- The Hartley transform has same computational complexity than the FFT, which allows for a fast evaluation of the neural operator.

**Weaknesses:**

- One of the main weaknesses of the paper is the motivation for the Hartley transform. The authors mention that the main limitation of the FNO is that it ``provides suboptimal solutions'' because the resulting functions can be complex-valued. However, I do not see any evidence supporting that this could be a limitation in practice as one could simply take the real part of the network since the output will likely have a small imaginary part if the loss function is small after training on real data.
- The authors only consider problems where the PDE is known so there is little motivation for using a PINO approach over PINN (or even simply a traditional numerical method like finite-difference or finite elements). The experiments must include a comparison against a PINN technique.
- The numerical experiments in the paper do not demonstrate that the proposed architecture improves the performance of the Fourier Neural Operator, as acknowledged by the authors for Burger's equation and observed in Fig. 1.

**Questions:**

- In the 2nd paragraph, what do the authors mean by suboptimal solutions?
- The last paragraph on p.3 has citations not rendered.
- Eqs. (5-7) should include the domain on which the PDE is defined, as well as a proper definition of the variables, and initial/boundary conditions.
- The loss functions in Eq. (8) should be mathematically defined.
- The discussion regarding the activation function in Section 3 requires references for claiming that GELU is preferred in physics-informed deep learning.
- In the last paragraph of section 3, it is inexact to state that the numerical scheme provides exact solution. What is the advantage of using the present approach over the numerical techniques, which are much faster and have convergence guarantees?
- In Figure 1, 4th column, why is the absolute error negative?
- Table 2: these results should be repeated over several training runs and contain deviation errors. Is it statistically significant to mention the ``reduced reconstruction loss" in the first paragraph of p. 7 if the difference only appears on the 4th digit?
- In the 2nd paragraph of the discussion, the author mention that their architecture achieve a ``more random trajectory'', this is a vague term which should be make precise.
- In the last sentence of the discussion: the authors mention that the activation function made no difference. I am wondering why it is discussed in the paper, which is about using the Hartley transform for neural operators.

---

> ### Author Response · Authors · 2023-11-21
> **We thank you for your candid and insightful review**
>
> Dear Reviewer,
>
> Thank you for your insightful feedback on our paper. We are grateful for the opportunity to enhance our work based on your suggestions. Though, given the undertaking that would necessitate a major revision, we will be withdrawing this submission in order to address all of these concerns before submitting to a future venue. We acknowledge the need for stronger evidence supporting the use of the Hartley transform and will address this in the revised manuscript. Additionally, we will include comparisons with PINN techniques and traditional numerical methods to better demonstrate the effectiveness of our approach.
>
> We will clarify the term "suboptimal solutions," ensure all citations are correctly rendered, and revise the mathematical definitions and domain specifications in Eqs. (5-7). References supporting our claim about the GELU activation function will also be added, as this was a rather large oversight. We will also fix the way absolute error is shown in Figure 1 and Table 2 and provide more solid statistical data from multiple training runs to show how the proposed method might possibly improve upon absolute error. We will make sure that the improvement is not small and is, in fact, statistically significant if we include this particular case as we revise the research. We will also show how this method is still novel and useful despite the convergence guarantees of state-of-the-art and well-known numerical methods.
>
> Finally, we will refine our discussion to better explain the architecture's features and the relevance of the activation function in our approach.
>
> Thank you once again for your constructive critique. We look forward to submitting our improved manuscript at a later time and venue.

---

### Official Review · Reviewer_uzwQ · 2023-10-29

**Soundness:** 2 fair
**Presentation:** 2 fair
**Contribution:** 1 poor
**Rating:** 3
**Confidence:** 3

**Summary:**

In this paper, an application of the Hartley transform to Physics-Informed Neural Operator (PINO) is discussed. The Fourier Neural Operator is one of the typical neural operators. This method is derived by replacing the Fourier transform used in this neural operator with the Hartley transform. The proposed method is applied to some specific problems and the performances of the methods are discussed.

**Strengths:**

When a Fourier transform is performed, complex numbers usually appear in the computation process. In contrast, the Haretley transform has the advantage that it can be computed using real numbers only, so the computation is expected to be accelerated.

**Weaknesses:**

The authors do not give any particular theoretical analysis for their proposed method. Therefore, the main contribution of this paper is considered to be numerical experiments. The authors do not give any particular theoretical analysis for their proposed method. Therefore, the main contribution of this paper is considered to be numerical experiments. However, the problem considered is limited to one-dimensional problems. Practically important multidimensional problems are not considered. I do not believe that this paper has made a sufficient contribution for publication.

**Questions:**

As mentioned in Future Work by the authors, I believe that application to multi-dimensional problems and the introduction of sophisticated architecture such as the AHNO attention mechanism are important and needed for publication. I encourage the authors to resubmit this paper after these improvements.

---

> ### Author Response · Authors · 2023-11-21
> **We thank you for your insightful and candid review.**
>
> Dear Reviewer,
>
> We would like to extend our utmost appreciation for the positive and helpful feedback provided on our paper. Thank you for recognizing the potential of our research, namely in its applicability to multi-dimensional challenges and the incorporation of advanced designs such as the AHNO attention mechanism.
>
> After doing a thorough study of the areas that have been identified as needing improvement, we have determined that it is necessary to withdraw our current proposal. Our proposed method for future submissions entails the meticulous incorporation of multidimensional applications and the AHNO attention mechanism within our framework. We believe that by implementing the suggested enhancements now, as you suggest, the work will be significantly improved. Thank you once again for your insightful feedback and encouragement. Your review has been instrumental in guiding our future work and ensuring the quality of our research.

---

### Official Review · Reviewer_sSA6 · 2023-10-30

**Soundness:** 3 good
**Presentation:** 3 good
**Contribution:** 3 good
**Rating:** 3
**Confidence:** 5

**Summary:**

This paper proposes a novel neural operator structure using the Hartley transform and suggests combining this neural operator structure with Physics-Informed Neural Networks (PINNs) to better incorporate the PDE loss. Overall, the idea is worth publishing; however, the paper's writing and formatting require substantial improvement. The comparisons in the experiments are also quite inadequate, hence I recommend the authors to revise the paper meticulously before submitting it to the next venue.

**Strengths:**

The integration of the Hartley transform with neural operators and PINNs is a novel idea proposed in this paper.

**Weaknesses:**

1. There is significant room for improvement in the paper's writing and formatting. Many reference links in the paper are represented as "?" symbols, tables in the paper are not centered, the best results are not highlighted, and there's an entire page (page 6) with a large figure without any accompanying text. These issues give the impression that the paper was hastily prepared to meet a deadline without sufficient revisions.
2. The paper provides scant explanations on why the integration with PINNs is essential, and what advantages does the Hartley transform-based Neural Operator have when combined with PINNs compared to the Physics-Informed Neural Operators (PINO)? Moreover, why is the use of complex numbers in Fourier neural operators considered a limitation? These essential inquiries regarding the necessity of the ideas presented were not well-addressed in the paper. Additionally, in many experimental results, we do not observe a significant improvement of the Hartley Neural Operator (HNO) over the Fourier Neural Operator (FNO). Despite the novelty of the idea, the paper has ample room for improvement in terms of writing and experimental validation.

**Questions:**

None

---

> ### Author Response · Authors · 2023-11-21
> **We thank you for your insightful and candid review.**
>
> Dear Reviewer,
>
> Thank you sincerely for your insightful and comprehensive review of our paper. We greatly appreciate the time and effort you have invested in evaluating our work. We will work to, above all, fix all minor typos and errors in our future submissions that, by themselves, detract from the presentation. Furthermore, we will include more details on the integration with PINNs, advantages of using Hartley transform-based neural operators, and identify settings where the computational disadvantage of FNOs can be clearly seen. Since this will be a major revision, we have decided to withdraw our paper from this conference. We plan to address each of your concerns meticulously in our future submission at a different venue. Rest assured, our next version will feature improved writing, formatting, and a more robust experimental validation to better convey the novelty and utility of our work. Your constructive criticism has provided us directions for improvement, and we are grateful for the opportunity to enhance our work.

---

### Official Review · Reviewer_6U54 · 2023-10-31

**Soundness:** 2 fair
**Presentation:** 1 poor
**Contribution:** 2 fair
**Rating:** 3
**Confidence:** 5

**Summary:**

The propose to use the Discrete Hartley Transform (HNO) rather than the FFT in the Fourier Neural Operator for a more efficient scheme. In particular, they use the HNO in a PINNs framework within PINO, which is a Neural Operator that adds the PDE to the loss function as a soft constraint. They study two PDEs, the linear parabolic and smooth heat equation and thermodynamic energy equation

**Strengths:**

- The authors study an important problem of improving deep learning models for solving PDEs, which are prevalent in science and engineering problems.
- It is nice that the authors consider adding PDE information into data-driven Neural Operator methods although this has already been covered with PINO (https://arxiv.org/abs/2111.03794) with mixed results.
- The main novelty is in proposing to use the Hartley Transform basis and its computational efficiency, which I'm not convinced is significant enough novelty.
- The authors identify and highlight an important problem with FNO and operators in general that is the curse of dimensionality. In particular, these methods are not scalable in 3D space and time since the resulting tensor is 6D and the operations are quite expensive. In fact, this real-world 3D case is the motivation for developing DL methods since the curse of dimensionality also impacts the state-of-the-art numerical methods.
- Nice use of numerical techniques such as the fourth order RK4 for the time-stepping rather than the commonly used first order Forward Euler.

**Weaknesses:**

- Very related work such as the Multi-wavelet Neural Operator (Gupta et. al, "Multiwavelet-based Operator Learning for Differential Equations", NeurIPS 2021) needs to be cited, which proposes using a multi-wavelet instead of Fourier basis and added as an additional baseline.
- The test cases of only 2 simple PDEs, e.g., heat and thermal is too simple and far less than the large number of test cases in FNO (Li et. al, https://arxiv.org/abs/2010.08895, ICLR 2021) and boundary-constrained NO (Saad et., al, https://arxiv.org/abs/2212.07477, ICLR 2023). In addition, Hansen et. al, "Learning Physical Models that Can Respect Conservation Laws", ICML 2023 provides a PDE benchmarking framework from "easy" to "hard" PDEs, where the heat equation studied here is a simple case. I would like the method tested on harder problems with shocks, e.g., Stefan in this paper or hyperbolic conservation laws.
- In addition Saad et. al, ICLR 2023 shows that adding the PINNs loss to Neural Operators can actually be more unstable than the data-driven training and this work should also be cited especially since the authors note their are issues with their proposed approach near the boundary. It seems like future work is needed to resolve this, such as the mixed results on Burgers' equation.
- Rather than starting the introduction with a focus on Neural Operators, I think providing background on PDEs, their importance in the scientific community and numerical methods would be good to add.
- Also references to other state-of-the-art SciML methods, such as MeshGraphNets (Pfaff et. al, https://arxiv.org/abs/2010.03409, ICLR 2021), MP-PDE (Brandstetter et. al, "Message Passing Neural PDE Solvers", ICLR 2022 (Spotlight)) are missing and the main focus of the literature review is Neural Operators.
- The novelty is in using the Hartley Transform basis and not the physics-informed part since that is applying PINNs out-of-the-box to Neural Operators as it was done with PINO.
- Problem definition section where the PDE is defined with initial condition and boundary conditions and the assumptions is missing.
- Some of the preliminaries on Discrete Hartley Transform, e.g., Eqn. 3 can be moved to an appendix, similarly with the long comparison to wavelets (where Gupta et. al should be cited) and discrete cosine transforms (this is interesting to further investigate.) and discussion of PINNs etc should be moved to background appendix sections.
- Listing viscous Burger's in Eqn. 5 seems out of place and should be moved to experiment section or a separate equations section and similarly for diffusion and themodynamic energy to make more room for the description of the proposed method.
- Several of the references seem quite dated.
- References have ? in "Operating on the Diffusion Equation, a key equation in Machine Learning. F"
- Ignoring the initial conditions " (since their GitHub implementation has this commented out in their code anyway)" is not an acceptable answer as to why they are ignored and much too informal in the writing.
- Paper organization is quite poor since there is no dedicated method section. It is mixed with background on these equations.
- The reference to PINO in discussion section is incorrect and should be Li et. al (https://arxiv.org/abs/2111.03794), 2021
- The plots are hard to read the title, x and y-axis labels and are not informative on the quality of the method.
- The work is a nice initial investigation but not ready for publication.

Minor
- Title is a bit long and simpler title may be better for the main message of the paper with too many keywords of Neural Operators and Physics-Informed Networks.
- Best not to start the introduction with bold paragraph header.
- Parenthesis around references is not consistent such as with FNO on the first line.
- Floating equations should have punctuation following it. Commas after equations are missing.
- "salable" instead of scalable at the end of the first paragraph.
- Matérn repeated in "Using GRF For Initial Conditions" paragraph

**Questions:**

- The method seems effective for PDEs with real-valued solutions but what happens with PDEs with complex-valued output? Will the method will work?
- Why are the authors arguing that the linear diffusion equation is key in ML? This is the simplest PDE that numerical methods can solve to high order.
- Is PINO compared to, i.e., FNO + PINNs loss? The experiments should compare to data-driven FNO and FNO + PINNs loss.
- How are the number of Hartley and Fourier layers chosen? It seems too hand-tuned and weakens the contributions isnce Fourier layers are still needed in some cases.

---

> ### Author Response · Authors · 2023-11-21
> **We thank you for your insightful and candid review.**
>
> Dear Reviewer,
>
> We are immensely grateful for your detailed and insightful review of our manuscript. Your thorough analysis and constructive comments are crucial in highlighting the strengths and areas for improvement in our submissions. We deeply appreciate your recognition of our efforts in advancing deep learning models for solving Partial Differential Equations (PDEs), a critical area in science and engineering. Your acknowledgment of our novel approach in integrating PDE information into data-driven neural operator methods and our exploration of the Hartley Transform basis is particularly encouraging.
>
> In light of your concerns and questions, we recognize the necessity of a more comprehensive and detailed approach in our next submission. Therefore, we have decided to withdraw our current manuscript and aim to resubmit it for an upcoming conference. In our revised version, we will address your queries regarding the efficacy of our method in solving PDEs with complex-valued outputs. We will also include more experimental setups in our comparison with respect to models such as FNO + PINNs loss, and more complex PDEs mentioned in our references. This will allow us to refine our manuscript further and ensure it meets the high standards of publication.
>
> We acknowledge the need to cite very related work, such as the Multi-wavelet Neural Operator, and will add this as an additional baseline. To address the concern about the simplicity of our test cases, we will expand our experiments to include more complex PDEs, including those with shocks and hyperbolic conservation laws. We will also cite relevant works such as Saad et. al, ICLR 2023, and others that you have mentioned to provide a broader and more accurate context for our research. We will start the introduction with a focus on PDEs, their importance in the scientific community, and numerical methods before delving into neural operators. We will also include references to other state-of-the-art SciML methods and ensure that our literature review is comprehensive and up-to-date. We will also include a problem definition section where the PDE is defined with initial conditions, boundary conditions, and assumptions. To improve the organization and readability of the paper, we will move some of the preliminary discussions to appendices and reorganize the sections to include a dedicated method section. We will revise our plots to make them more readable and informative.
>
> Finally, we will answer your questions about whether our method works for PDEs with complex-valued output, why we chose to focus on the linear diffusion equation, and how to exactly compare PINO loss to FNO + PINNs loss. The selection of Hartley and Fourier layers will also be explained in more detail to demonstrate the robustness of our approach.
>
> Once again, thank you for your invaluable feedback. We are committed to addressing these points comprehensively in our continuing work on this before considering a new submission.